# Nanotechnological Formulation Incorporating *Pectis brevipedunculata* (Asteraceae) Essential Oil: An Ecofriendly Approach for Leishmanicidal and Anti-Inflammatory Therapy

**DOI:** 10.3390/polym17030379

**Published:** 2025-01-30

**Authors:** Estela Mesquita Marques, Lucas George Santos Andrade, Luciana Magalhães Rebelo Alencar, Erick Rafael Dias Rates, Rachel Melo Ribeiro, Rafael Cardoso Carvalho, Glécilla Colombelli de Souza Nunes, Daniele Stéfanie Sara Lopes Lera-Nonose, Maria Julia Schiavon Gonçalves, Maria Valdrinez Campana Lonardoni, Melissa Pires Souza, Emmanoel Vilaça Costa, Renato Sonchini Gonçalves

**Affiliations:** 1Laboratory of Chemistry of Natural Products, Department of Chemistry, Federal University of Maranhão (UFMA), São Luís 65080-805, Brazil; estela.marques@discente.ufma.br (E.M.M.); lucas.gsa@discente.ufma.br (L.G.S.A.); 2Laboratory of Biophysics and Nanosystems, Department of Physics, Federal University of Maranhão, São Luís 65080-805, Brazil; luciana.alencar@ufma.br (L.M.R.A.); erick.rates@discente.ufma.br (E.R.D.R.); 3Graduate Program in Health Sciences, Federal University of Maranhão (UFMA), São Luís 65080-805, Brazil; melo.rachel@ufma.br (R.M.R.); carvalho.rafael@ufma.br (R.C.C.); 4Research Nucleus in Pharmaceutical Sciences Program, State University of Maringá, Maringá 87020-900, Brazil; gcsnunes2@uem.br; 5Department of Clinical Analysis and Biomedicine, State University of Maringá (UEM), Maringá 87020-900, Brazil; dssllnonose2@uem.br (D.S.S.L.L.-N.); pg405366@uem.br (M.J.S.G.); mvclonrdoni@uem.br (M.V.C.L.); 6Postgraduate Program in Chemistry, Federal University of Amazonas (UFAM), Manaus 69080-900, Brazil; melissa.souza@ufam.edu.br (M.P.S.); evc@ufam.edu.br (E.V.C.); 7Department of Chemistry, Federal University of Amazonas (UFAM), Manaus 69080-900, Brazil

**Keywords:** ecofriendly nanogel, leishmanicidal activity, anti-inflammatory effects

## Abstract

Cutaneous leishmaniasis caused by *Leishmania amazonensis* is a significant public health issue. This study aimed to evaluate an ecofriendly, thermosensitive nanogel, developed using a low-energy, solvent-free method, incorporating F127 and Carbopol 974P copolymers, and enriched with *Pectis brevipedunculata* essential oil (EO*Pb*) for its leishmanicidal and anti-inflammatory properties. The nanogel was prepared and characterized through FTIR, DLS, SEM, and AFM to confirm the incorporation of EO*Pb* as well as its stability and rheological properties. In vitro leishmanicidal activity was evaluated on *Leishmania amazonensis* promastigotes, and in vivo anti-inflammatory effects were assessed using a rat paw edema model. In vitro, nGF3 (EO*Pb*-loaded nanogel) demonstrated significant leishmanicidal activity, with promastigote mortality rates exceeding 80% at 24 h and 90% at 48 h. In vivo, nGF1, nGF2, and nGF3 exhibited anti-inflammatory effects, with nGF2 and nGF3 reducing edema by 62.7% at 2 h post-treatment. The empty nanogel (nGF0) showed minimal anti-inflammatory activity. The ecofriendly EO*Pb*-loaded nanogel (nGF3) demonstrated strong leishmanicidal and anti-inflammatory effects, presenting a promising candidate for cutaneous leishmaniasis treatment. Further studies are necessary to explore its clinical potential.

## 1. Introduction

Leishmaniasis represents a significant global health challenge, affecting millions in diverse geographical regions, particularly in tropical and subtropical areas [1,2]. This parasitic disease, caused by various species of Leishmania and transmitted by sandflies, manifests itself in cutaneous [3], mucocutaneous [4], and visceral forms [5], each presenting distinct clinical complexities and health burdens [6]. Although often neglected within the broader scope of public health initiatives, leishmaniasis remains endemic in more than 90 countries, with an estimated 700,000 to 1 million new cases each year. Current control measures are limited by socioeconomic factors, challenges in vector control, and the lack of an effective vaccine, making the disease’s persistence a critical concern. Addressing the multidimensional aspects of leishmaniasis demands a coordinated global response, integrating scientific, medical, and socio-political strategies to reduce its impact on affected populations [7].

Current treatments for leishmaniasis primarily rely on a limited array of drugs, including antimonials, amphotericin B, miltefosine, and paromomycin [8]. These treatments, while effective, pose several challenges: many have significant toxicity, require prolonged administration, and often demand hospitalization, especially in cases requiring intravenous medication [9,10]. The efficacy of these drugs also varies by Leishmania species and geographical location, complicating treatment protocols. Furthermore, drug resistance is emerging in some regions, exacerbating treatment difficulties and underscoring the need for alternative therapies [11]. For public health systems, particularly in endemic areas, the financial burden of leishmaniasis treatment is substantial. High drug costs, combined with the need for infrastructure to support safe administration and follow-up care, strain limited healthcare budgets, often redirecting resources from other critical areas. This economic strain highlights the urgent need for investment in the development of affordable, effective, and accessible therapies that can alleviate the financial and operational pressures on public health systems worldwide [12].

Natural products, particularly essential oils (EOs) from Amazon rainforest species, are gaining attention as promising alternatives for sustainable leishmaniasis treatments. Rich in terpenoids, phenolic compounds, and other bioactive molecules, these EOs exhibit strong antimicrobial and antiparasitic properties [13,14]. Recent studies have focused on incorporating EOs into nanotechnological delivery systems, such as nanoparticles and liposomes, to enhance therapeutic efficacy by improving drug stability, targeting, bioavailability, and reducing toxicity [15,16,17,18,19,20]. This approach not only offers potential for more effective leishmaniasis therapies but also promotes the sustainable use of Amazonian biodiversity, aligning conservation with innovation in global health [21].

Nanogels are an innovative and effective approach for treating cutaneous leishmaniasis (CL) lesions, combining high loading capacity and controlled release of active compounds, making them ideal carriers for EOs. EOs exhibit anti-inflammatory, antimicrobial, and wound-healing properties, which reduce inflammation, accelerate healing, and lower the parasitic burden. Topical application of EO-loaded nanogels ensures targeted and sustained delivery, enhancing therapeutic efficacy and minimizing systemic side effects [22,23]. These systems improve antimicrobial, anti-inflammatory, and regenerative properties, promoting skin repair while reducing infection risks [24,25,26,27,28]. Despite their promise, studies on EO-loaded nanogels focusing on leishmanicidal and anti-inflammatory activities are limited, emphasizing the need for further research to optimize formulations and evaluate their efficacy in CL treatment [29,30].

This study explores the antimicrobial potential of *Pectis brevipedunculata* (Asteraceae) essential oil (EO*Pb*) loaded in F127/974P nanogels. *Pectis brevipedunculata*, native to tropical and subtropical regions of the Americas, especially Brazil, Argentina, and Uruguay, is known for its therapeutic properties, including anti-inflammatory and antimicrobial effects [31]. Its EO, rich in monoterpenes like neral, geranial, α-pinene, and limonene, exhibit strong antimicrobial activity, making the plant a promising source of natural antimicrobial agents [32,33]. Our previous research demonstrated the larvicidal activity of EO*Pb*-loaded nanogels against *Aedes aegypti* larvae, achieving effective inactivation without cytotoxicity [34]. Based on these results, the current study evaluates the in vitro activity of the nanogel against *Leishmania amazonensis* promastigotes (*LLa*) and its in vivo anti-inflammatory effects in a model of carrageenan-induced rat paw edema.

These findings suggest that the nanogel (nGF) shows strong potential as a multifunctional nanocarrier. It demonstrates time- and concentration-dependent efficacy against Leishmania amazonensis promastigotes, with significant potency even at lower concentrations. Mortality rates of 38.15% at 24 h and 33.03% at 48 h were observed, with the cytotoxic effect attributed to the OE*Pb* composition in the gel matrix. Additionally, in vivo studies showed that nanogel application at OE*Pb* concentrations of 0.25%, 0.50%, and 1% achieved maximum efficacy 2 h after inflammatory agent inoculation, with edema reductions of 44.1%, 62.7%, and 54.4%, respectively. This ecofriendly nanogel, developed through a low-energy, solvent-free process, presents a sustainable and innovative approach to wound treatment that offers potential as an effective therapeutic solution for CL. By leveraging the potent bioactive properties of EO*Pb* within a biodegradable gel matrix, this formulation aligns environmental responsibility with therapeutic efficacy, positioning it as a promising alternative for sustainable, targeted treatment in the management of leishmaniasis-related skin lesions.

## 2. Materials and Methods

### 2.1. Materials

Pluronic F127 (poly(ethylene oxide)-poly(propylene oxide)-poly(ethylene oxide) triblock copolymer (MW = 12,600 g/mol; (EO_99_PO_67_EO_99_)), ultrapure water, anhydrous sodium sulfate (≥99%), sodium chloride (≥99%), deuterated chloroform (CDCl_3_), alkane standard mixture XTT (2,3-bis(2-methoxy-4-nitro-5-sulfophenyl)-5-[(phenylamino)carbonyl]-2H-tetrazolium hydroxide), PMS (N-methyl dibenzopyrazine methyl sulfate), penicillin, streptomycin, fetal bovine serum, and amphotericin B were commercially acquired from the Merck company (Rahway, NJ, USA). Carbopol 974P NF polymer was provided by IMCD Brasil (São Paulo, Brazil).

### 2.2. Plant Material

*P. brevipedunculata* (*Pb*) was collected from the Universidade Federal do Maranhão (UFMA) campus in São Luís, Brazil (2°33′20.5″ S, 44°18′32.7″ W). A voucher specimen (No. 5287) was deposited at the Rosa Mochel Herbarium (SLUI), Universidade Estadual do Maranhão (UEMA), São Luís, Brazil. The plant collection adhered to the Brazilian biodiversity protection regulations, registered under the SisGen code AAFB38B.

### 2.3. Extraction Procedure

The essential oil from *Pb* (EO*Pb*) was extracted through hydrodistillation using a Clevenger-type apparatus following the methodology described in a previous study [34]. Air-dried *Pb* (300 g) was cut into small pieces using pruning shears to optimize extraction efficiency. The plant material was then combined with 500 mL of distilled water in a flask and hydrodistilled for 3 h following reflux onset. Post-extraction, the oil/water mixture was centrifuged at 3500 rpm for 10 min at 25 °C, using a clinical centrifuge Centribio 80-2B, analog (São Paulo, Brazil). The remaining water was removed by treating the mixture with anhydrous sodium sulfate, yielding a final EO*Pb* yield of 0.80% based on initial plant material weight (Figure 1).

### 2.4. GC–MS and NMR Analyses of EO*Pb*

The OE*Pb* analyses were conducted utilizing GC–MS and NMR techniques, following previously established methodologies [34]. Briefly, the GC and GC-MS analyses were performed using Shimadzu systems, employing a fused capillary column (RXi-1MS) with helium carrier gas. The temperature program consisted of optimized settings, with 10 mg/mL samples in CH_2_Cl_2_ injected at a 1:50 split ratio. Retention indices were determined using n-alkane standards, and peak areas and retention times were measured to calculate relative amounts. GC–MS analyses utilized a Shimadzu QP2010 SE system with AOC-20i auto-injector, employing identical conditions to GC. EO components were identified by comparing retention times, indices, and MS spectra with standards and data obtained from the literature (ADAMS and FFNSC libraries). NMR spectra (^1^H, ^13^C, and DEPT ^13^C) were acquired on a BRUKER Avance III HD spectrometer (11.75 Tesla, 500.13 MHz and 125.76 MHz). The samples were dissolved in deuterated chloroform (CDCl_3_) with chemical shifts reported in ppm relative to tetramethylsilane (TMS) as internal reference.

### 2.5. Preparation of Nanogel Formulations and Stability Evaluation

The nanogel formulations were developed according to a previously published paper [34]. F127 copolymer (20% *w*/*w*) was gradually introduced into cold distilled water maintained in an ice bath (5–10 °C), with gentle stirring to facilitate hydration of each copolymer flake. Subsequently, 974P (0. 2% *w*/*w*) was added incrementally and the solution was stirred gently at 5–10 °C until complete dissolution was achieved. Subsequently, EO*Pb* was added dropwise to the mixture while continuously stirring for 30 min. The solution was refrigerated overnight at 5 °C to ensure complete solubility of all components (Figure 1). The formulations were labeled according to the OE*Pb* concentration (% *w*/*w*): nGF0 (prepared without OE*Pb*), nGF1 (containing 0.25%), nGF2 (containing 0.50%), and nGF3 (containing 1.0%). To assess the impact of temperature on the physical and chemical stability of nanogel formulations, accelerated stability tests and shelf-life tests were conducted according to the methodology described in the previously published work [34], following ANVISA and US Pharmacopeia guidelines. Chemical stability was analyzed using GC-MS by comparing EO*Pb* chromatograph profiles over a 28-day period.

### 2.6. FTIR Analysis

FTIR analyses were performed in reflectance mode using a Shimadzu Tracer-100 FTIR spectrophotometer (Kyoto, Japan). Nanogel samples were freeze-dried and compressed into pellets with KBr, while pure OE*Pb* was analyzed using Attenuated Total Reflectance (ATR) mode. The spectrometer was equipped with a horizontal ATR accessory, featuring a ZnSe crystal window (PIKE Technologies) for ATR-FTIR measurements. The spectra were collected within the 400–4000 cm^−1^ range, with a resolution of 8 cm^−1^ and 50 scans. For sample preparation, the material was evenly applied to the ATR crystal surface. After each spectrum was acquired, the crystal window was thoroughly cleaned with hexane and acetone before proceeding with further measurements. The UV-Vis absorption spectra of the nanogels, prepared at a concentration of 5.0 × 10^−3^ g/mL in water, were recorded at room temperature using a Shimadzu UV-Vis 1800 spectrophotometer (Kyoto, Japan).

### 2.7. DLS Analysis

Average hydrodynamic diameter (D_*h*_) analyses were determined by dynamic light scattering (DLS) in water at 25 °C and with a 40 mW semiconductor laser of 658 nm with a Litesizer 500 (Anton Paar GmbH) instrument (Module BM 10). The D_*h*_ measurement was performed using a quartz cuvette of 3.0 mL. The ζ potential was performed using a low-volume cuvette (Univette). All the measurements were performed in triplicate (mean ± SD).

### 2.8. Scanning Electron Microscopy (SEM)

The morphology of nGF0 and nGF3 was examined using SEM analysis. The samples were initially frozen in liquid nitrogen at −196 °C and then lyophilized for 24 h using a Thermo Micro Modulyo freeze dryer (Thermo Electron Corporation, Pittsburgh, PA, USA). After lyophilization, the samples were coated with a thin layer of metal using a BAL-TEC SCD 050 Sputter Coater (Balzers, Liechtenstein), and their morphology was analyzed at magnifications of 100× and 50× using an FEI Quanta 250 microscope (Thermo Fisher Scientific, Karlsruhe, Germany).

### 2.9. Atomic Force Microscopy (AFM)

The morphology of nGF0 and nGF3 was also examined using AFM analysis. All samples were used after the freeze-drying process. All samples were examined following the lyophilization process. A Multimode 8 microscope (Bruker, Santa Barbara, CA, USA) was used for the analysis, operating in PeakForce Quantitative Nanomechanics (QNM) mode. In this mode, the probe oscillates at 1 kHz, below its resonance frequency, acquiring force curves at every oscillation cycle. The probes used in this study had a nominal spring constant of 0.4 N/m and a tip radius of 2 nm (specific to the probe model), with a scanning resolution of 256 × 256 lines and a scan rate of 0.5 Hz per acquired map. Simultaneously with the force curve acquisition, nanomechanical properties such as Young’s modulus and adhesion were also measured [35]. At one point on the topographic map acquired for the sample nGF3, an actual probe–sample interaction curve (Figure 2B) can be identified, showing both the approach curve (blue) and retraction curve (red).

The separation between these two curves indicates energy dissipation during the approach–retraction cycle. During the approach phase (Index 1 on the curve), when the probe is far from the sample surface, attractive van der Waals forces act until the probe undergoes a “jump-to-contact”, making contact with the sample surface (Index 2 on the curve). During the indentation phase, the linear regime of the interaction (Index 3) is used to calculate Young’s modulus based on the Derjaguin–Muller–Toporov (DMT) model. For the interaction between a non-deformed conical probe and a rigid sample surface, this model is expressed by Equation (1) [36]:(1)$$F\left(\delta \right)=\frac{4}{3}\cdot \left(\frac{E}{1-\nu }\right)\cdot \sqrt{R}\cdot \delta^{3/2}$$
where *E* is Young’s modulus, ν is Poisson’s ratio, δ is the indentation depth, and *R* is the probe radius. Additionally, during the retraction cycle of the curve shown in Figure 2B, the probe–sample separation encounters resistance, manifested as a downward deflection of the cantilever due to attractive forces (Index 4 on the curve). The maximum resistance observed during probe detachment from the sample surface characterizes adhesion [37,38]. After overcoming this resistance, the probe moves away from the sample surface, completing the acquisition of a single force curve (Indices 5 and 6 on the curve).

### 2.10. In Vitro Assay Against Promastigotes of Leishmania (Leishmania) Amazonensis (*LLa*)

*LLa* (strain PH8) cells were cultured in 199 culture medium, supplemented with 10% inactivated fetal bovine serum and antibiotics (100 UI/mL penicillin and 0.1 mg/mL streptomycin). The cultures were incubated at 27 °C with constant subculturing. In 96-well plates, the culture of *LLa* promastigotes was seeded in RPMI 1640 medium, resulting in a final concentration of $2\times 10^{7}$ leishmania/mL after the addition of the compounds. The nanogels were diluted in RPMI 1640 medium and added at concentrations ranging from 2.2 to 0.7 mg/mL in OEPb. The plates were then incubated at 27 °C for 24, 48, and 72 h. The viability of the promastigotes was determined using the colorimetric XTT method. A mixture containing 20% of XTT:PMS and 60% of NaCl solution (at concentration of the 0.9%) was added to the plates. The plates were then incubated for 4 h at 37 °C with 5% CO_2_. After this period, absorbance was measured using a spectrophotometer with filters set to 450–620 nm, and the percentage of inhibition was estimated by comparison with untreated cells. Amphotericin B was used as a positive control. The experiments were conducted in triplicate and in the absence of light. The percentage of mortality was calculated based on a logarithmic regression of the control curve made only with Leishmania and the culture medium, starting with a concentration of $2\times 10^{7}$ leishmania/mL and diluted in a ratio of two, down to a concentration of 6.25 × 10^5^ leishmania/mL.

### 2.11. In Vivo Evaluation of the Anti-Inflammatory Activity of Nanogels

To characterize the anti-inflammatory effect of the produced nanogels, the carrageenan-induced paw edema experimental protocol was performed in adult male Swiss mice, following the methodology of Sulaiman et al. (2010) with modifications. Approved by the Ethics Committee of CEP-UFMA (Research Ethics Committee of AGEUFMA), under approval code 23115.038817/2024-44, on 1 May 2024. Topical formulations nGF0–nGF3 were evaluated against diclofenac sodium gel 10 mg/g, which was selected as the standard anti-inflammatory topical drug. The animals were divided into six groups (n = 5), as follows: Control, which received NaCl 0.9% saline solution; Groups II, III, and IV, which received nGF1, nGF2, and nGF3, respectively; Group V received nGF0; and Group VI received diclofenac sodium gel 10 mg/g. After 1 h, acute inflammation was induced in the right hind paws of the animals by intraplantar injection of 0.05 mL of carrageenan (1% *w*/*v*). A digital caliper was utilized to measure the increase in paw thickness (Ct) immediately following the carrageenan injection (0 h) and then every hour for 4 h thereafter. Any increase in paw thickness was considered an indicator of inflammation (Figure 3). The calculated inflammation inhibition is expressed as(2)$$\mathrm{Edema}\;\mathrm{inhibition}\;\left(\%\right)=\left(\frac{\left[{\left(C_{t}-C_{0}\right)}_{\mathrm{Control}\;\mathrm{group}}-{\left(C_{t}-C_{0}\right)}_{\mathrm{Treated}\;\mathrm{group}}\right]}{{\left(C_{t}-C_{0}\right)}_{\mathrm{Control}\;\mathrm{group}}}\right)\times 100$$
where $C_{t}$ is the paw measurement after carrageenan treatment at time *t*, and $C_{0}$ is the initial (basal) paw measurement.

### 2.12. Statistical Analysis

For the statistical analysis, data were evaluated using a single-criterion approach through ANOVA, followed by Tukey’s post-test for pairwise comparisons and Kruskal–Wallis with Dunn’s post hoc test for non-parametric data. Statistical significance was determined based on predefined criteria. All statistical analyses and graphical representations were performed using Prism 9 software (GraphPad, San Diego, CA, USA). Values with statistically significant differences were explicitly highlighted.

## 3. Results and Discussion

### 3.1. Development of the Nanogels

According to our previous publication, the chemical composition of OE*Pb* was analyzed using GC-MS, identifying thirteen components that accounted for 91.17% of the oil’s composition. The major component was citral, comprising 64.58% of the oil, represented by two isomeric oxygenated monoterpenes: geranial (36.06%) and neral (28.52%). Other significant constituents included α-pinene (15.73%) and limonene (8.28%). The chemical structures of the major compounds were confirmed through ^1^H and ^13^C NMR [34]. As stated in our previous publication, a variety of combinations of F127 and 974P percentages were evaluated to identify the ideal formulation that would enable thermosensible behavior in the nanogel. Combinations of 5–10:0.1–0.3% did not exhibit a sol–gel transition, remaining liquid at both 5 °C and 30 °C. In contrast, the 20:0.1–0.3% combinations stayed liquid at 5 °C and transitioned to a semi-solid state at 30 °C, with the 20:0.2% formulation being optimized for a rapid sol-to-gel transition. This combination allowed for the incorporation of EO*Pb*, and formulations with up to 1% of EO*Pb* were stable, transparent, and free from phase separation, even after accelerated stability tests. In contrast, the nanogel at EO*Pb* concentration above 1% led to instability and phase separation. Thus, the formulations nGF1-nGF3 (Figure 1F) that proved to be stable after accelerated stability tests and shelf-life assays and exhibited thermosensitive behavior with a rapid sol–gel transition were selected for this study.

### 3.2. Spectroscopy Characterization of the Nanogels

For the spectroscopic characterization assays, the nanogel containing the maximum percentage of EO*Pb* (nGF3) was selected. In our previous study, we demonstrated that the chemical stability of nGF3 could be assessed through GC analysis, revealing a high similarity between the chromatographic profiles of EO*Pb* and nGF3. EO*Pb* and nGF3 [34] both showed the same elution order of α-pinene, limonene, geranial, and neral, ensuring that the composition of EO*Pb* is preserved even after the nanoencapsulation process within the F127/974P polymeric matrix.

The FTIR of the EO*Pb* was performed to support the GC–MS and NMR characterization, confirming the encapsulation of EO*Pb* on the F127/974P matrix. In the FTIR spectrum of EO*Pb* (Figure 4A), the main absorption bands observed confirm the presence of four key compounds, validating the GC–MS and NMR characterization (Table 1). The absorption bands between 2954 and 2920 cm^−1^ correspond to the symmetric and asymmetric stretching of C–H bonds associated with aldehyde and hydrocarbon groups. The bands at 1442 and 1377 cm^−1^ indicate C–H bending vibrations of alkane groups, while the high-intensity band at 1674 cm^−1^ and the low-intensity band between 887 and 789 cm^−1^ are associated with C=C bonds in trisubstituted alkenes. The small shoulder at 1712 cm^−1^ corresponds to the C=O stretching of conjugated aldehyde groups. In the FTIR of nGF3, we observed a strong and sharp absorption band at 3676 cm^−1^, which resulted from a pronounced redshift with Δν = 112 cm^−1^ from the band at 3564 cm^−1^, observed in the empty nanogel nGF0 spectrum (Figure 4B). These findings suggest that a reduction in energy is necessary for the stretching of O–H bonds. This is likely attributable to the ability of EO*Pb* molecules to interfere with some interactions within the F127/974P blend, allowing them to integrate into the nGF3 matrix. The absorption band observed at 3384 cm^−1^ in nGF0 shifted to 3340 cm^−1^, reflecting a blue shift of 44 cm^−1^. This change may be indicative of the establishment of new intermolecular hydrogen bonds between the F127/974P blend and the predominant components of EO*Pb*, suggesting that the more hydrophobic EO*Pb* molecules are preferentially situated within the PPO chains of F127. This interpretation is supported by the heightened energy needed for the anti-symmetric stretching of C–H bonds, which is evidenced by the blue shift to 19 cm^−1^ when we compared the absorption band at 2904 cm^−1^ in nGF0 with that at 2885 cm^−1^ in the nGF3 spectra. This change indicates a significant enhancement in the hydrophobic interactions within the system, which is further supported by the morphological changes observed in the nGF0 to nGF3 nanogels, as demonstrated below through SEM and AFM analyses.

The size of nGF0 and nGF3 were quantitatively assessed through DLS analysis (Table 2). This technique has become essential for evaluating the size distribution of nanoparticles in solution. By examining the variations in scattered light intensity due to Brownian motion, DLS offers critical insights into the D_*h*_ and PDI of nanogels, which are vital for determining the homogeneity and stability of the formulations. DLS measurements were carried out in solution systems, diluted to a concentration below 1% (*w*/*v*). Under these conditions, which are below the critical micellar concentration (CMC) of F127, the demicellization process is favored, resulting in a decrease in the number of smaller particles. This phenomenon occurs because the total entropy of the system allows smaller particles to aggregate into larger particles, resulting in a (D_*h*_) of 661.03 ± 6.1 nm.

When 1% of EO*Pb* is solubilized in the F127/974P polymeric matrix, the (D_*h*_) reduces to 30.44 ± 12.1 nm, indicating a wide particle dispersion within the solution. In accordance with the morphological analysis, the EO*Pb* significantly alters the morphology of the nanogel, leading to a well-defined polymeric organization, with an average size of 40.58 ± 7.98, which is relatively smaller than those calculated for the micellar domains in nGF0. Nanoparticles aimed for drug delivery typically range from 10 to 200 nm in size; thus, selecting the optimal size is critical for enhancing the efficacy of drug delivery systems. It is well established that F127 tends to self-assemble into spherical aggregates in dilute solutions (CMC = 0.34 mM) [39]. This size range is optimal because nanoparticles of this dimension demonstrate favorable traits, such as increased cellular uptake, extended circulation times in the bloodstream, and improved biodistribution [40].

In a prior study by our group, we demonstrated the optimization process of F127/974P proportions, which resulted in a formulation with thermoresponsive behavior [34]. This observation highlights that micellar domains are not only important for stabilizing the nanogel structure but are also essential for achieving an efficient sol–gel equilibrium. Although the morphological analyses of nGF3 revealed non-spherical domains due to the incorporation of OEPb, the particle sizes found were close to 40 nm; and during the initial stages of nanogel preparation, we hypothesized that F127 underwent micelle formation, leading to the coexistence of micellar domains and planar domains [16,17]. This coexistence is critical, as micellar domains play a key role in maintaining structural stability and facilitating sol–gel transitions, particularly at higher concentrations of F127 (approximately 10–15% *w*/*w*) [41].

The ζ potential measurements for nGF0 and nGF3 were found to be −0.2 and −1.6 mV, respectively. Although the observed negative values were not high, they were sufficient to maintain the nanogel systems with good stability throughout the accelerated stability tests. Additionally, the incorporation of 1% w/w of OE*Pb* resulted in a more negative ζ potential value, which can be inferred as an increase in the stability of the nanometric particles in solution when compared to the empty nanogel system, nGF0.

### 3.3. Morphological Characterization of the Nanogels

The SEM technique surged as an indispensable tool for unraveling the intricate landscapes of nano- and microstructured formulations. This technique, well known for its high-resolution imaging capabilities, is crucial in characterizing materials at scales that are often significant in pharmaceutical formulations for a comprehensive understanding of their properties and potential applications. SEM analysis allows for examining the surface and cross-sectional morphology of the freeze-dried nGF0 and nGF3 formulations. The SEM micrographs of nGF0 (Figure 5) show that the F127 and 974P polymer blend has a morphological pattern consisting of the porous nature of the matrix with interconnected channel-like structures characteristic of the cross-linked arrangement of 974P polymer chains, as a result of intermolecular hydrogen bonding. SEM images also allow us to observe that a non-uniform surface layer covers the internal porous regions (Figure 5B,C). These morphological characteristics can help explain the properties of nGF0. The porous distribution can be filled by the liquid phase, causing the viscosity of nGF0 to be high due to strong hydrogen bonds with the water.

However, the addition of 1% EO*Pb* at nGF0 significantly alters the morphology of nanoformulation. The SEM micrographs of nGF3 show well-defined planar regions with porous absence in all analyzed materials (Figure 6). As observed by FTIR analysis, the strong intermolecular interaction between EO*Pb* and PEO polymer chains prevents the formation of pores during the freeze-dried process. Furthermore, the cross-sectional morphology of the nGF3 shows thick planes organized in layers (Figure 7). These morphological characteristics support the physical properties of nGF3. The strong interaction between the polymer and EO*Pb* can help to explain its high viscosity, and the well-defined structural organization of the layers allows for easy movement of the planes, which is reflected macroscopically in a material with soft texture and high spreadability.

AFM analyses were employed to obtain morphological details of the nGF0 and nGF3 formulations. AFM provided high-resolution images of the gel structures at the nanoscale, offering insights into the particle size, shape, and surface characteristics such as Young’s modulus and adhesion force. The topographic maps presented in Figure 8 reveal detailed structural characteristics of the nGF0 formulation. Two distinct domains are indicated by the yellow and green arrows Figure 8A. The domain indicated by the yellow arrow corresponds to a relatively flat region with an average height of 5.92 ± 3.00 nm (n = 15), resulting from strong intermolecular interactions between the F127 and 974P polymers. On the other hand, the domain indicated by the green arrows shows relatively spherical geometries, with an average height of 108.74 ± 19.41 nm (n = 19) and an average diameter of 279.09 ± 38.93 nm (n = 20), indicating the formation of F127 micelles (Figure 8A,B).

As observed through FTIR, the F127 micelles are stabilized by cross-linked hydrogen bonds with the 974P chains. The presence of these distinct domains suggests the contribution of van der Waals forces during the formation of the F127/974P polymer blend, leading to the formation of a highly stable structural network, with height values and geometries consistent with the expected characteristics based on the properties of the individual components. The topographic analysis of the nGF0 nanogel also reveals the presence of a third domain, characterized by a micellar cubic structure, as illustrated in Figure 8C. This observation is intriguing as it suggests the formation of a distinct phase coexisting with the spherical and flat domains. Studies previously reported in the literature have demonstrated the polymorphic behavior of amphiphilic block copolymers at concentrations above 5% by mass. Gel phases with diverse structures, including lamellar, normal and reverse hexagonal, normal and reverse bicontinuous cubic, and normal and reverse micellar cubic phases, have been documented in the literature, supporting our findings [41,42,43].

Figure 9A,B detail the Young’s modulus values for the domains of the nGF0 system shown in the previous micrographs. The measurements of Young’s modulus provide insights into the mechanical properties of the structural domains. It is observed that both the relatively spherical micellar structures and the cubic forms of F127 predominantly exhibit lower Young’s modulus values. In contrast, the flat domains, resulting from the strong intermolecular interactions between F127 and 974P, display relatively higher values. These findings are consistent with observations that the F127/974P domains form more rigid flat structures, characterized by brighter regions in Young’s modulus maps. Meanwhile, the micellar structures of F127, in the absence of OE*Pb*, are empty, contributing to a greater fluidity and softness of these domains, as evidenced by the darker regions in the maps.

Adhesion force analyses stem from a combination of four distinct interactions between the AFM probe and the sample surface: van der Waals, electrostatic, capillary, and chemical forces [37,38]. It is essential to delineate the individual contributions of each interaction to the overall adhesion phenomenon. Consistency in probe material composition, geometric parameters (such as tip radius and shape), and environmental conditions (including temperature and relative humidity) is imperative to ensure reproducibility and comparability of the analyses. This standardization enables the elucidation of how variations in adhesion contrast among samples reflect underlying heterogeneities in surface composition. Additionally, the geometric characteristics of the probe, particularly its conical shape, play a significant role in minimizing contact angle and consequently reducing the influence of capillary forces [38]. Systematic investigation of adhesion forces is crucial to gain comprehensive insights into the complex intermolecular interactions that govern the AFM probe–sample interactions, providing additional support for the interpretation that the domains exerting the highest adhesion forces on the AFM probe result from the strong hydrogen bonds between F127 and 974P (Figure 9C,D). In contrast, the central regions of the F127 micelles result in weak adhesion forces, as a response to the dipole and induced dipole interactions (Keesom and Debye forces) between the PPO hydrophobic chains and the AFM probe. Additionally, the micellar structures exhibit moderately clear outlines with median adhesion force values, corroborating the interpretation that these structures are stabilized by hydrogen bonds cross-linked with the planar domains of F127/974P.

The AFM micrographs acquired for the nanogel nGF3 reveal a distinct morphological pattern compared to the nGF0 material, clearly attributed to the incorporation of 1% (*w*/*w*) of EO*Pb*. The topographic maps of nGF3 (Figure 10A,C) indicate that the regions marked with red arrows are probably associated with the EO molecules incorporated on the surface of the nanogel, forming well-defined and organized layered structures with an average height of 7.39 ± 0.79 nm (n = 17). The absence of spherical and cubic geometry domains in the nGF3 material suggests a significant restructuring of the micellar architecture driven by the incorporation of EO*Pb*(indicated by the green arrows in Figure 10C,D). According to FTIR analyses, the incorporation of EO*Pb* into the nGF0 matrix significantly increases the intensity and energy of the band associated with the C—H vibrational modes, suggesting a strong interaction between EO*Pb* molecules and the PPO chains of the F127 micelles, as well as the pores formed by the intermolecular interactions of F127–974P. Small areas with an average height of 1.37 ± 0.21 nm (n = 20) and an average size of 40.58 ± 7.98 nm (n = 20) are observed, which are relatively smaller than those calculated for the micellar domains of nGF0.

Young’s modulus analyses for the nanogel nGF3 support the elucidation of its mechanical properties. Regions with higher concentrations of EO*Pb* reveal lower Young’s modulus values compared to regions with lower concentrations, confirming the emollient characteristic of EO*Pb* (Figure 11A,B). The adhesion force maps shown in Figure 8A,B corroborate these characteristics, as regions with higher concentrations of EO*Pb* exhibit lower adhesion force values due to weaker van der Waals interactions (Figure 11C,D).

### 3.4. In Vitro Leishmanicidal Activity of Nanogel

The results of our in vitro investigation into the leishmanicidal activity of the nanogel formulations are presented in Figure 12A. Initially, the empty nanogel (nGF0) was evaluated to assess the toxicity of the nanoformulation excipients in the leishmanicidal response, which did not exhibit any antileishmanial activity against promastigotes of *LLa*. This result confirms the inert nature of the vehicle (i.e., only the formulation excipients), as it does not induce any leishmanicidal activity. In contrast, nGF3 exhibited substantial antileishmanial activity across all concentrations tested at both 24 and 48 h. Notably, at the highest concentration (2.2 to 0.28 mg/mL in EO*Pb*), nGF3 achieved a promastigote mortality rate exceeding 80% at 24 h, with an increase to over 90% at 48 h. These findings suggest a time- and concentration-dependent efficacy, with nGF3 demonstrating significant potency against *LLa*. At lower concentration, nGF3 still maintained a notable level of activity, achieving mortality rates of 46 and 68% at 24 and 48 h, respectively. Thus, it is evident that the cytotoxic potential of the nGF3 nanogel is attributed to the OE*Pb* composition within the gel matrix. Figure 12B presents representative images of Leishmania cultures treated with the nanogel formulation (nGF3) and OEPb at varying concentrations, highlighting the morphological changes observed over 24 and 48 h. The pronounced antileishmanial effects observed for nGF3 highlight its potential as a topical treatment option for cutaneous leishmaniasis. Given the high rates of promastigote mortality achieved, particularly at prolonged exposure, nGF3 presents itself as a promising candidate for further investigation. This efficacy supports the possibility of developing an effective localized therapy that may reduce systemic exposure and associated toxicity. Further studies to elucidate the mechanism of action, along with in vivo testing, are warranted to advance the development of nGF3 as a therapeutic agent for cutaneous leishmaniasis treatment.

### 3.5. In Vivo Evaluation of the Anti-Inflammatory Potential of Nanogels

The results of the anti-inflammatory activity of the nanogels are presented in Figure 13. Upon analysis, it was observed that treatments with the nanogels, namely nGF1, nGF2, and nGF3, exhibited peak efficacy at 2 h post-inoculation of the inflammatory agent, resulting in edema reductions of 44.1 ± 5.8%, 62.7 ± 4.4%, and 54.4 ± 6.7%, respectively. The results demonstrated that the nanogels possess statistically significant anti-inflammatory potential. While their efficacy did not entirely match that of the standard treatment, diclofenac sodium gel at 10 mg/g (65.5 ± 5.5%), the development of an ecofriendly nanogel with notable anti-inflammatory activity highlights its promise as a sustainable alternative with effective therapeutic potential. However, this anti-inflammatory effect was not sustained in subsequent time points for any of the formulations or the standard drug. Notably, the graphical data indicate a relatively diminished effect for nGF1 compared to the other groups, which included diclofenac sodium, nGF2, and nGF3. Additionally, treatment with nGF0 exhibited low edema inhibition at the 2 h mark (14.9 ± 4.7%), with a significant statistical difference observed between this group and the other treatments (*p* < 0.001). Collectively, these findings suggest that the nanogels at the three concentrations analyzed possess anti-inflammatory action that is independent of dosage; however, further studies are warranted, particularly for nGF2 and nGF3. The transient nature of the observed effects indicates that while these formulations demonstrate promise, their efficacy may be limited in duration, highlighting the need for optimization in formulation or delivery methods. Future investigations should explore the underlying mechanisms of action, as well as the potential for sustained release or combination therapies that could enhance their anti-inflammatory properties over extended periods.

## 4. Conclusions

This study demonstrated that the ecofriendly nanogels loaded with EO*Pb* exhibit significant potential as multifunctional therapeutic systems for the treatment of CL. The nanoformulations displayed high leishmanicidal efficacy in vitro, achieving mortality rates above 90% within 48 h. The in vivo results demonstrated a maximum edema reduction of 62.7% within two hours of treatment, suggesting that the evaluated nanogels possess potential as anti-inflammatory agents and could be considered a promising alternative to standard drugs such as diclofenac. Spectroscopic and morphological characterizations confirmed the chemical stability and structural integrity of the nanogels, preserving the bioactive composition of EO*Pb* after the nanoencapsulation process. Future studies should explore the mechanical aspects underlying the leishmanicidal and anti-inflammatory effects of the nanogels, as well as conduct clinical trials and field tests to validate their effectiveness in real-world settings. Furthermore, investigating the broader therapeutic applications of these nanogels could significantly contribute to sustainable and targeted strategies for combating leishmaniasis and other neglected diseases. The integration of nanotechnological systems like nGF3 underscores the importance of Amazonian natural products in pharmaceutical innovation, advancing environmentally responsible and effective solutions to global health challenges.

## Figures and Tables

**Figure 1 polymers-17-00379-f001:**
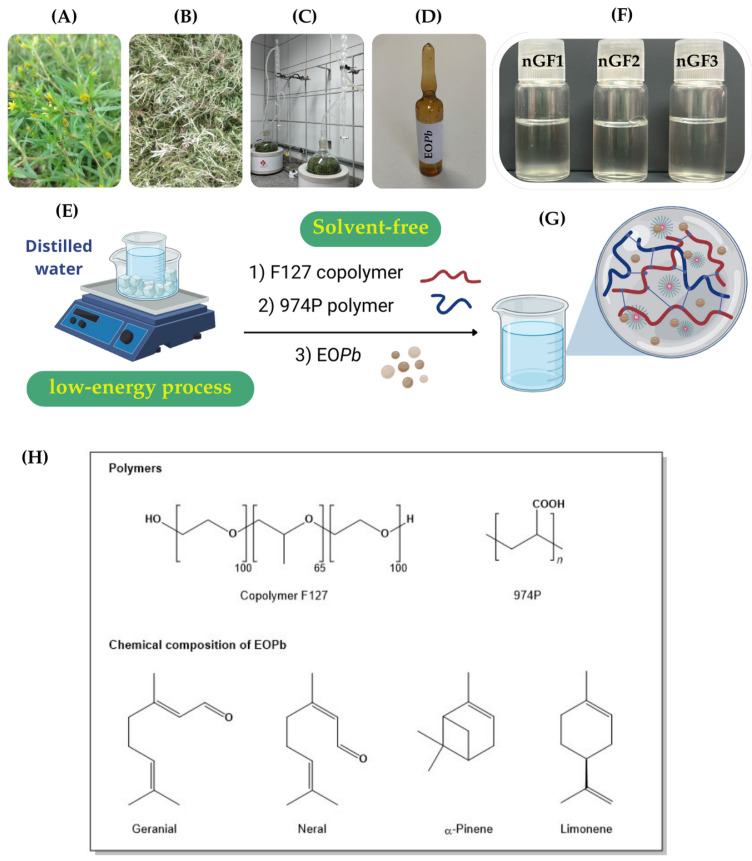
Experimental sequence for the OE*Pb* extraction procedure: Collection of *Pb* (**A**), air drying of the plant material (**B**), grinding and hydrodistillation under controlled conditions (**C**), followed by drying and storage of OE*Pb* in a sealed amber vial (**D**). The nanogel preparation methodology was performed using a low-energy, solvent-free procedure (**E**). Photograph of the nanogels nGF1–nGF3 (**F**), along with a schematic representation of the structural organization of the F127/974P polymer blend and OE*Pb*-loading F127 micelles (**G**). Chemical structure of the polymers used in the preparation of the nanogels and chemical composition of the major chemical constituents of EO*Pb* (**H**).

**Figure 2 polymers-17-00379-f002:**
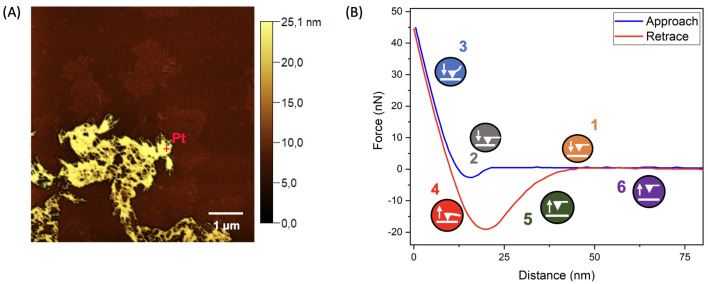
AFM force curves. (**A**) Topographic map of the sample, displaying distinct contrasts for higher regions (brighter areas) and lower regions (darker areas). Point 1 (Pt 1) corresponds to the specific region where the force curve was acquired. (**B**) Representative force curve obtained from Point 1 in image A, illustrating the approach (blue) and retraction (red) cycles. The indices along the curve highlight the distinct stages of the probe–sample interaction, as detailed in the graphical insets: (1) the probe is distant from the sample surface, (2) the probe establishes contact with the sample surface, (3) the linear regime of the curve during indentation, (4) the maximum resistance observed during probe–sample separation, (5) the probe retracts from the sample, and (6) the approach–retraction cycle concludes.

**Figure 3 polymers-17-00379-f003:**
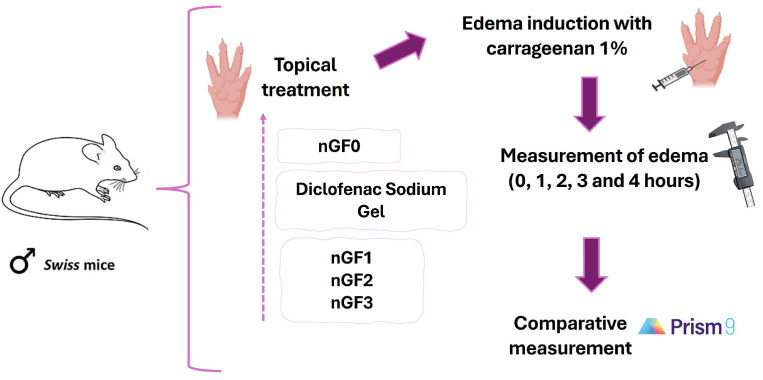
Experimental sequence for in vivo evaluation of the anti-inflammatory potential of nanogels.

**Figure 4 polymers-17-00379-f004:**
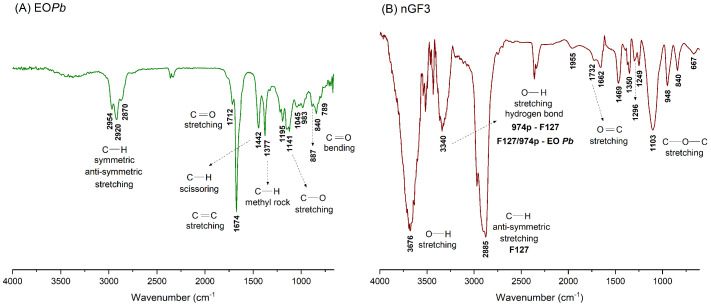
FTIR spectra: (**A**) Essential oil of *Pectis brevipedunculata* (EO*Pb*) and (**B**) nanogel formulation containing 1% *w*/*w* EO*Pb* (nGF3).

**Figure 5 polymers-17-00379-f005:**
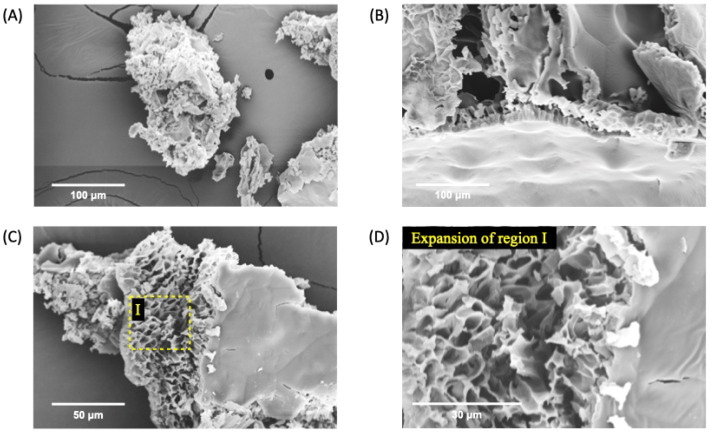
SEM micrographs of nGF0 at a magnification of 1000× (**A**), 2000× (**B**,**C**), and 5000× (**D**) after the freeze-drying process.

**Figure 6 polymers-17-00379-f006:**
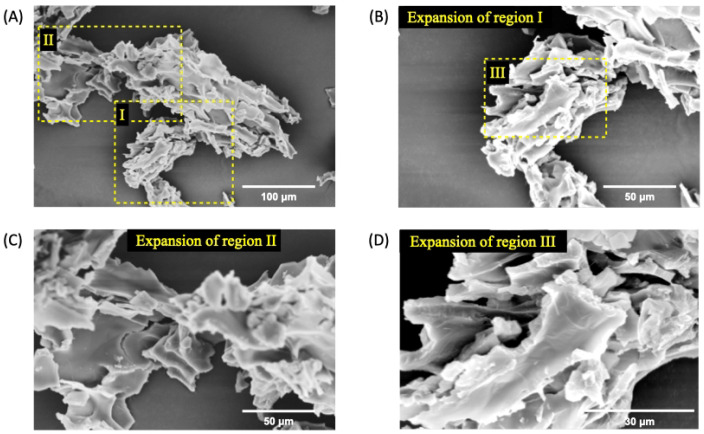
SEM micrographs of nGF3 at a magnification of 500× (**A**), 1000× (**B**), 2000× (**C**), and 5000× (**D**) after the freeze-drying process. The SEM micrographs show well-defined planar regions with the absence of pores.

**Figure 7 polymers-17-00379-f007:**
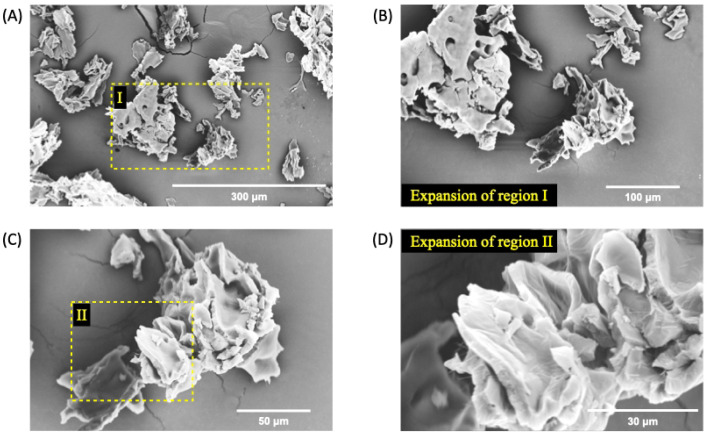
SEM micrographs of nGF3 at a magnification of 1000× (**A**), 2000× (**B**,**C**), and 5000× (**D**) after the freeze-dying process. The SEM micrographs show thick planes organized in layers.

**Figure 8 polymers-17-00379-f008:**
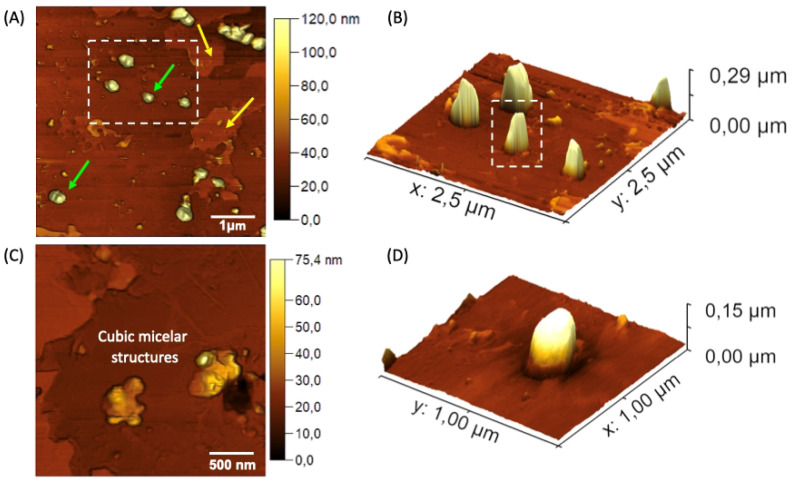
AFM topographic maps of the nGF0 nanogel. (**A**) Yellow arrows indicate spherical structures formed by F127 micelles, with an average height of 108.74 ± 19.41 nm (n = 19), and green arrows indicate flat regions with an average height of 5.92 ± 3.00 nm (n = 15). (**B**,**D**) Expanded 3D micrographs showing the spherical structures in detail. (**C**) Cubic micellar structures formed by aggregational processes of F127 micelles.

**Figure 9 polymers-17-00379-f009:**
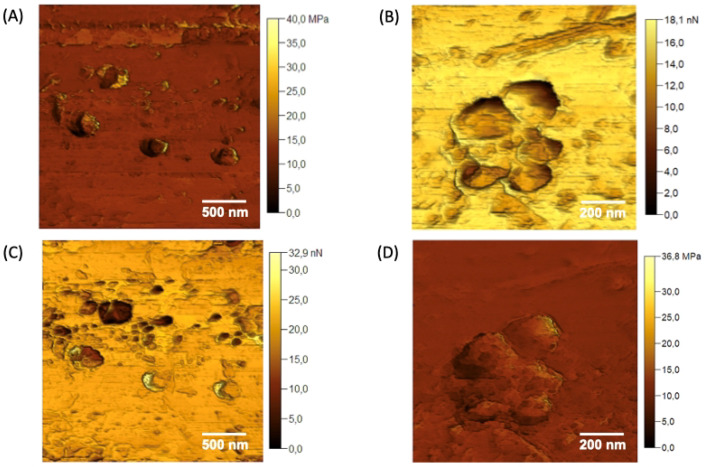
Young’s modulus maps (**A**,**B**) and adhesion force maps (**C**,**D**) were acquired for the different domains observed in the nGF0.

**Figure 10 polymers-17-00379-f010:**
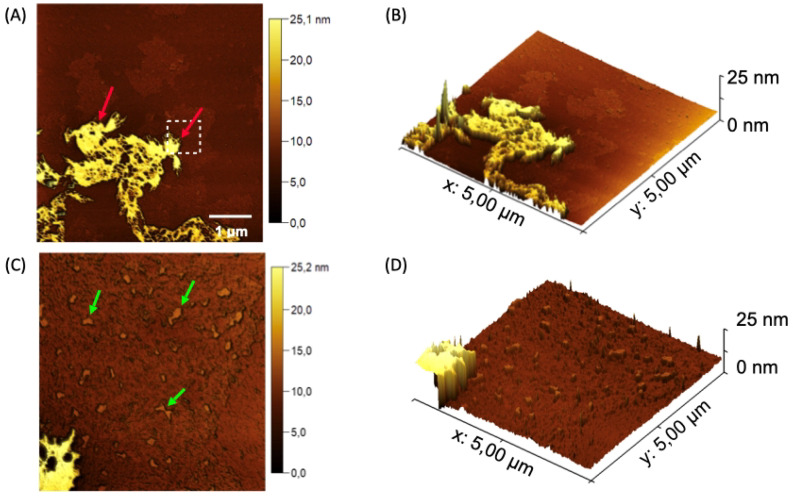
Topographical AFM maps of the nanogel nGF3. (**A**) Red arrows indicate flat structures formed by the presence of EO*Pb* on the surface of the nGF0 matrix, with an average height of 7.39 ± 0.79 nm (n = 17). (**B**) Green arrows indicate the incorporation of EO*Pb* into the F127 micellar structures and the pores of the nGF0 material, with average height values of 1.37 ± 0.21 nm (n = 20) and size of 40.58 ± 7.98 nm (n = 20). (**C**,**D**) 3D micrographs showing the flat nGF3 structures.

**Figure 11 polymers-17-00379-f011:**
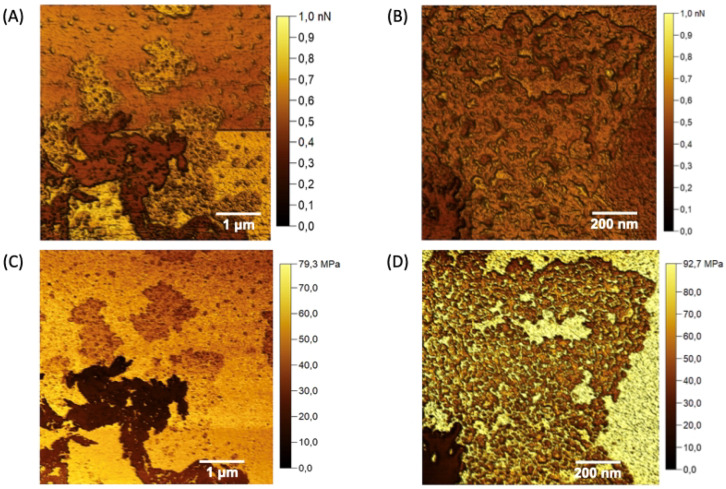
Maps of Young’s modulus (**A**,**B**) and adhesion forces (**C**,**D**) acquired for the nanogel nGF3.

**Figure 12 polymers-17-00379-f012:**
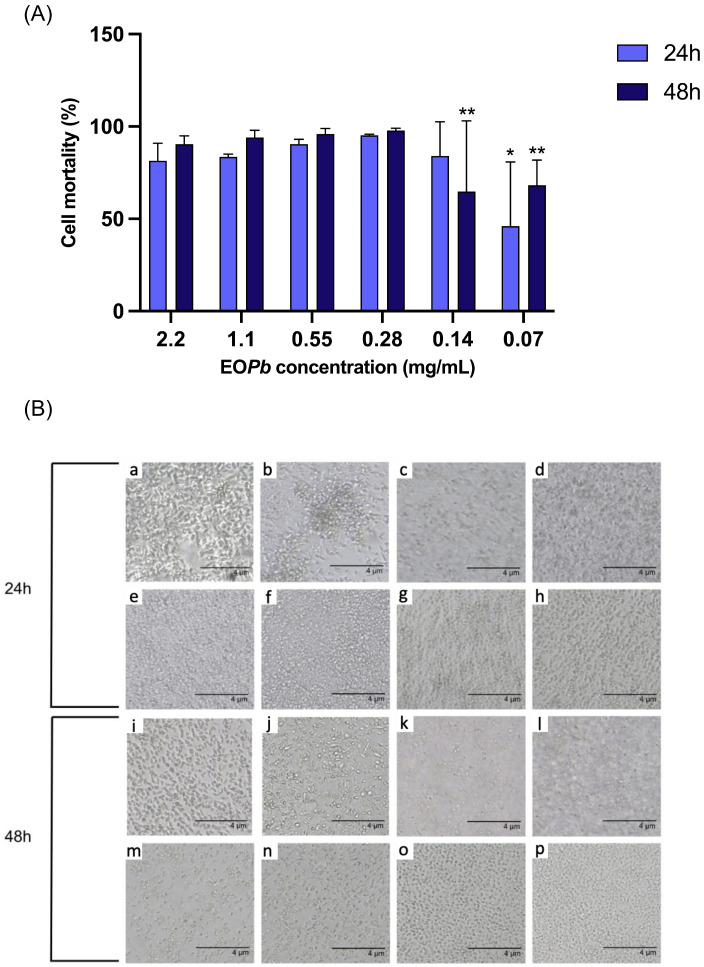
(**A**) In vitro leishmanicidal activity of nanogel formulation (nGF3) demonstrating concentration- and time-dependent efficacy against *LLa* (* *p* < 0.05, ** *p* < 0.01, Kruskal–Wallis with Dunn’s post hoc test). (**B**) Representative images of *Leishmania* cultures treated with the nanogel formulation (nGF3): (a) Untreated culture after 24 h. (b) Culture treated with the nanogel (nGF0) for 24 h. (c–h) Cultures treated with OE*Pb* at concentrations of 2.2, 1.1, 0.55, 0.28, 0.14, and 0.07 mg/mL, respectively, for 24 h. (i) Untreated culture after 48 h. (j) Culture treated with nGF0 for 48 h. (k–p) Cultures treated with OE*Pb* at concentrations of 2.2, 1.1, 0.55, 0.28, 0.14, and 0.07 mg/mL, respectively, for 48 h.

**Figure 13 polymers-17-00379-f013:**
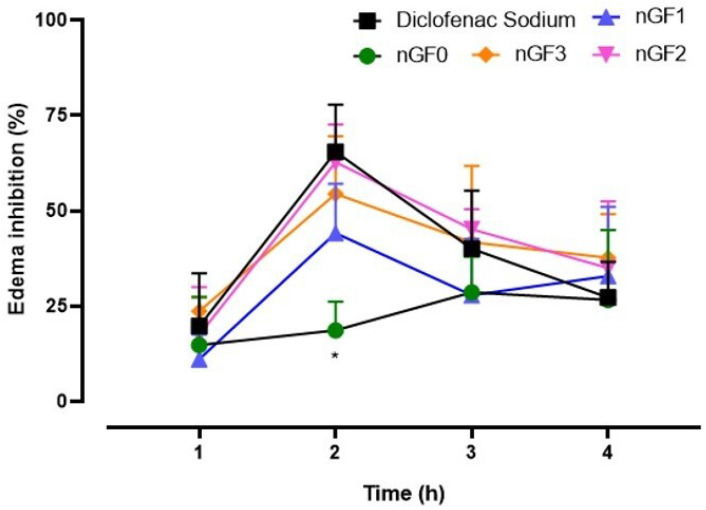
Topical anti-inflammatory action of nanogels in an experimental paw edema model in mice (* *p* < 0.05, ANOVA, Tukey’s test).

**Table 1 polymers-17-00379-t001:** Infrared spectroscopy data for EO*Pb* and nGF3 compounds.

Compound	Wavenumber (cm^−1^)	Assignment	Compound Class
**EO** * **Pb** *	2954	C—H stretching	Alkene
	2920	C—H stretching	Alkane
	2870	C—H stretching	Aldehyde
	1712	C=O stretching	Aldehyde
	1674	C=C stretching	Alkene
	1442	C—H scissoring	Alkane
	1377	C—H rock	Alkane
	1141	C—O stretching	Aldehyde
	840	C=O bending	Aldehyde
**nGF3**	3676	O—H stretching	Alcohol
	3340	O—H stretching	Hydrogen bond
	1732	C=O stretching	Aldehyde
	1103	C—O—H stretching	Ether

**Table 2 polymers-17-00379-t002:** Particle size measurements (nm) of nGF0 and nGF3 nanogels obtained using DLS and AFM techniques, expressed as D_*h*_ ± standard deviation (SD), PDI, height (H) and diameter (D) ± SD.

Measurement	nGF0	nGF3
DLS: D_*h*_ ± SD; PDI	661.03 ± 6.1; 0.34	30.44 ± 12.1; 0.54
DLS: ζ potential (mV)	−0.2	−1.6
AFM: D ± DS (nm)	279.09 ± 38.93	40.58 ± 7.98
AFM: H ± DS (nm)	108.74 ± 19.41	1.37 ± 0.21

## Data Availability

Data are contained within the article.

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
