# Peer review of "Nanotechnological Formulation Incorporating Pectis brevipedunculata (Asteraceae) Essential Oil: An Ecofriendly Approach for Leishmanicidal and Anti-Inflammatory Therapy"

_polymers, 2025, doi:10.3390/polym17030379_

Round 1
Reviewer 1 Report
Comments and Suggestions for Authors
Please check the attached file for minor comments

Author Response
We would like to express our sincere gratitude to the reviewers for their insightful comments regarding our work, polymers-3424962. It is a pleasure to have the expertise of the reviewers, who will be instrumental in enhancing the quality of the final version of the manuscript. Your valuable feedback is greatly appreciated and will significantly contribute to the improvement of our research and its reception by readers. Please note that the corrections have been highlighted in red in the text. Additionally, a file containing detailed responses to all the reviewers' comments has been attached in PDF format for your convenience.

Reviewer 2 Report
Comments and Suggestions for Authors
This paper describes the synthesis of nanogel particles loaded with Pectis brevipedunculata essential oil and their application in leishmanicidal and anti-inflammatory therapy. While this study includes interesting results and has the potential to contribute significantly to the medical field, particularly in drug delivery systems, several concerns should be addressed before publication. My comments are as follows:
- Chemical Structures: The chemical structures of Pectis brevipedunculata and Carbopol 974P are unclear. The authors should provide these chemical structures to facilitate a better understanding of the interactions between Pectis brevipedunculata and the copolymer matrix in the nanogels.
- Dynamic Light Scattering (DLS) Analysis: The manuscript lacks visual data regarding the DLS analysis. The authors should include spectra illustrating the variation in scattering intensity distribution to help readers visually understand how the distribution changes with increasing concentrations of Pectis brevipedunculata essential oil (EOPb). Furthermore, they should present the variation in the normalized time correlation function of the scattering field, G₁(τ), over time to demonstrate its linear decay, which would confirm the formation of spherical particles. This point is particularly important as the SEM and AFM images of the nanogels do not clearly display a spherical morphology.
- Particle Size Optimization: On page 9, lines 328–330, the authors state that the nanogel particles have an optimal size for drug delivery. However, it is essential to optimize particle size to match the pore size of the target cells for effective application in drug delivery systems. The authors should clarify how the particle size can be manipulated to achieve this optimization.
By addressing these issues, the manuscript will provide a clearer understanding for readers and significantly enhance its overall quality.
Author Response
We would like to express our sincere gratitude to the reviewers for their insightful comments regarding our work, polymers-3424962. It is a pleasure to have the expertise of the reviewers, who will be instrumental in enhancing the quality of the final version of the manuscript. Your valuable feedback is greatly appreciated and will significantly contribute to the improvement of our research and its reception by readers. Please note that the corrections have been highlighted in blue in the text. Additionally, a file containing detailed responses to all the reviewers' comments has been attached in PDF format for your convenience.

Reviewer 3 Report
Comments and Suggestions for Authors
This manuscript provides details of fabrications and characterizations for nano gels containing Pectis brevipedunculata essential oil (EOPb) for the leishmanicidal and anti-inflammatory effects. Sufficient information on experimental procedures of FTIR, dynamic light scattering (DLS), SEM, AFM, in-vivo, and in-vivo tests were explained. Overall, the manuscript is quite comprehensive. There are still some technical issues that need to be addressed to improve the quality of the report.
1. Please provide the spectra of FTIR along with Table 1. The identified peaks should be marked on the FTIR spectra also. I noticed there is “Fig. S8”, which could be in the previously published study. The authors can add this citation to the spectra of FTIR.
2. Zeta potential values may be better to add to Table 2.
3. It may be better to provide examples of AFM curves for samples under the study (maybe a specific point). This will demonstrate how the Derjaguin-Muller-Toporov (DMT) model is used to determine Yong’s modulus. Note that the AFM loading-unloading curve should have an indentation phase and a detachment phase.
4. It is may be more illustrative if the optical images of in-vitro cultured promastigotes of LLa can be provided. I guess there are some stained cell images that could be available from the authors’ experiments.
5. I’m not as confident as the authors on the efficacy of the 3 nanogels of three concentrations for their anti-inflammatory action shown in the in vivo Evaluation of the Anti-Inflammatory Potential of Nanogels. Fig. 11 shows that the standard treatment, diclofenac sodium gel at 10 mg/g (65.5 ± 5.5%) is effective as compared to the newly designed nanogels with essential oil. I strongly suggest that the authors to change the tone in the whole manuscript by looking for “an alternative but equivalently effective anti-inflammatory nature intergradient of ……”.
Author Response
We would like to express our sincere gratitude to the reviewers for their insightful comments regarding our work, polymers-3424962. It is a pleasure to have the expertise of the reviewers, who will be instrumental in enhancing the quality of the final version of the manuscript. Your valuable feedback is greatly appreciated and will significantly contribute to the improvement of our research and its reception by readers. Please note that the corrections have been highlighted in purple in the text. Additionally, a file containing detailed responses to all the reviewers' comments has been attached in PDF format for your convenience.

Round 2
Reviewer 3 Report
Comments and Suggestions for Authors
The authors already replied and revised all comments raised in the first round of review. In particular, the inclusion of the AFM curve and in-vitro cell images are all necessary to provide a better illustration of the experimental results. There are no further comments on my part.
Comments on the Quality of English LanguagePlease check for typos or errors before the final submission.